# Exploring Importance and Regulation of Autophagy in Cancer Stem Cells and Stem Cell-Based Therapies

**DOI:** 10.3390/cells13110958

**Published:** 2024-06-01

**Authors:** Md Ataur Rahman, Ehsanul Hoque Apu, S. M Rakib-Uz-Zaman, Somdeepa Chakraborti, Sujay Kumar Bhajan, Shakila Afroz Taleb, Mushfiq H. Shaikh, Maroua Jalouli, Abdel Halim Harrath, Bonglee Kim

**Affiliations:** 1Department of Neurology, University of Michigan, Ann Arbor, MI 48109, USA; 2Global Biotechnology and Biomedical Research Network (GBBRN), Department of Biotechnology and Genetic Engineering, Faculty of Biological Sciences, Islamic University, Kushtia 7003, Bangladesh; 3Department of Biomedical Sciences, College of Dental Medicine, Lincoln Memorial University, Knoxville, TN 37923, USA; hoqueapu.ehsanul@gmail.com; 4DeBusk College of Osteopathic Medicine, Lincoln Memorial University, Harrogate, TN 37752, USA; 5Division of Hematology and Oncology, Department of Internal Medicine, Michigan Medicine, University of Michigan, Ann Arbor, MI 48109, USA; 6Department of Biological Sciences, University of Delaware, Newark, DE 19716, USA; rakibsust07@gmail.com (S.M.R.-U.-Z.); somdeepa7@gmail.com (S.C.); 7Biotechnology Program, Department of Mathematics and Natural Sciences, School of Data and Sciences, BRAC University, Dhaka 1212, Bangladesh; 8Department of Biotechnology and Genetic Engineering, Bangabandhu Sheikh Mujibur Rahman Science & Technology University, Gopalganj 8100, Bangladesh; sujaybge@gmail.com; 9Department of Internal Medicine, Yale School of Medicine, Yale University, New Haven, CT 06510, USA; shakila.taleb@yale.edu; 10Department of Otolaryngology—Head and Neck Surgery, Western University, London, ON N6A 4V2, Canada; mushfiqddc@gmail.com; 11Department of Biology, College of Science, Imam Mohammad Ibn Saud Islamic University (IMSIU), Riyadh 11623, Saudi Arabia; mejalouli@imamu.edu.sa; 12Zoology Department, College of Science, King Saud University, Riyadh 11451, Saudi Arabia; hharrath@ksu.edu.sa; 13Department of Pathology, College of Korean Medicine, Kyung Hee University, 1-5 Hoegidong, Dongdaemun-gu, Seoul 02447, Republic of Korea; 14Korean Medicine-Based Drug Repositioning Cancer Research Center, College of Korean Medicine, Kyung Hee University, Seoul 02447, Republic of Korea

**Keywords:** autophagy, cancer stem cells, stem cell therapy, cancer cell, organoid and tissue regeneration, therapeutic target

## Abstract

Autophagy is a globally conserved cellular activity that plays a critical role in maintaining cellular homeostasis through the breakdown and recycling of cellular constituents. In recent years, there has been much emphasis given to its complex role in cancer stem cells (CSCs) and stem cell treatment. This study examines the molecular processes that support autophagy and how it is regulated in the context of CSCs and stem cell treatment. Although autophagy plays a dual role in the management of CSCs, affecting their removal as well as their maintenance, the intricate interaction between the several signaling channels that control cellular survival and death as part of the molecular mechanism of autophagy has not been well elucidated. Given that CSCs have a role in the development, progression, and resistance to treatment of tumors, it is imperative to comprehend their biological activities. CSCs are important for cancer biology because they also show a tissue regeneration model that helps with organoid regeneration. In other words, the manipulation of autophagy is a viable therapeutic approach in the treatment of cancer and stem cell therapy. Both synthetic and natural substances that target autophagy pathways have demonstrated promise in improving stem cell-based therapies and eliminating CSCs. Nevertheless, there are difficulties associated with the limitations of autophagy in CSC regulation, including resistance mechanisms and off-target effects. Thus, the regulation of autophagy offers a versatile strategy for focusing on CSCs and enhancing the results of stem cell therapy. Therefore, understanding the complex interactions between autophagy and CSC biology would be essential for creating therapeutic treatments that work in both regenerative medicine and cancer treatment.

## 1. Introduction

Autophagy is a highly conserved cellular mechanism that plays a vital role in sustaining cellular homeostasis by breaking down malfunctioning organelles as well as proteins [1]. Conversely, cancer stem cells (CSCs), which are a specific type of cancer cells that possess the ability to regenerate and develop tumors, have attracted considerable interest due to their involvement in the start, advancement, and resistance to treatment of tumors [2]. Furthermore, the emergence of stem cell therapy has presented encouraging opportunities in the field of regenerative medicine and the treatment of cancer [3]. The comprehension of autophagy modulation within the framework of CSCs and stem cell therapy presents significant opportunities for the advancement of cancer therapies and regenerative medicine [4]. Autophagy is a highly regulated biological process characterized by the generation of autophagosomes, which are responsible for engulfing cytoplasmic cargo to facilitate lysosomal breakdown [5]. ATG proteins, which are crucial molecular components, serve a pivotal role in coordinating multiple phases of autophagy, encompassing the initiation, elongation, and maturation of autophagosomes [6]. The involvement of autophagy dysregulation has been associated with a range of pathological diseases such as cancer. The complex interactions among autophagy, CSCs, and stem cell treatment have become a central focus in cancer research in recent years.

Cancer stem cells (CSCs) are irregular tumor cells that replicate normal stem cells. CSCs can survive chemotherapy and radiotherapy, and they can also induce cancer recurrence and metastasis because they are resistant multi- or monopotent stem cells [7]. These cells contribute to tumor heterogeneity, metastasis, and disease recurrence, creating important hurdles for cancer treatment. Understanding the biological role of CSCs is critical for designing tailored therapeutic techniques aimed at eliminating this robust cell type [8]. Emerging research has revealed that CSCs have extraordinary flexibility and can aid in tissue regeneration and repair [9]. The CSCs’ ability to develop into numerous cell lineages highlights their role in tissue regeneration and homeostasis [10]. However, abnormal activation of CSCs can cause pathological situations such as cancer development and progression [11]. Organoids, three-dimensional in vitro models that mimic organ structure and function, have since emerged as effective tools for understanding CSC biology and tissue regeneration [12,13]. CSCs play an important role in organoid generation and maintenance, providing insights into organ development, disease modeling, and drug discovery [14].

Autophagy plays a dual role in CSC maintenance, functioning as both a pro-survival mechanism and a tumor suppressor route. While basal autophagy supports CSC self-renewal and treatment resistance, increased autophagy activation can result in CSC depletion and reduced tumor development [15]. Understanding the context-dependent regulation of autophagy in CSCs is critical for developing successful therapeutic approaches. Exploring the therapeutic potential of autophagy modulation holds promise for targeting CSCs and enhancing stem cell therapies [16]. Natural substances and synthetic medications that target autophagy pathways have been demonstrated to effectively suppress CSC growth, induce differentiation, and sensitize CSCs to conventional therapy [17,18]. Despite the therapeutic potential of autophagy regulation, there are significant hurdles and limitations to targeting autophagy in cancer treatment. Off-target effects, drug resistance, and potential negative effects on normal stem cells necessitate additional research to optimize the therapeutic techniques targeting autophagy in CSCs and stem cell therapies [19]. It may be possible to overcome therapeutic resistance, enhance treatment results, and use stem cells’ capacity for regenerative processes to promote tissue repair and regeneration by focusing on the autophagy pathways in CSCs [20]. Therefore, this review focuses on gaining a comprehensive understanding of the complex relationship among autophagy, cancer stem cells, and stem cell therapy and how this presents promising opportunities for the progression of cancer therapeutics and regenerative medicine.

## 2. Molecular Mechanism of Autophagy

The activation of autophagy involves several membranes associated with dynamic sequential events. Over 40 autophagy related (ATG) proteins are necessary for the various stages of autophagy [21]. Autophagy is triggered by various cellular stimuli such as nutrient fluctuations, phosphorylation/dephosphorylation, and extracellular or intracellular stimuli [22]. These stimuli lead to the formation of phagophores, a process initiated by the sequestration of cytoplasmic components [23]. The formation of a phagophore involves two protein complexes, namely a class III PI3 K Vps34 complex, consisting of Vps15, Vps34, Atg14, and Beclin1, and an Unc-51-like kinase 1 (ULK1) complex, comprising Atg13, Atg101, FIP200, and the serine/threonine kinase ULK1 [24]. The kinase depends on the presence of two additional autophagy proteins, specifically Atg13 or Atg8, along with Atg17 [25]. During a nutrient deprivation event, the altered ATP/AMP ratio is detected by AMPK, which then suppresses the mTOR activity, leading to the activation of the ULK1 complex as the mTOR can inhibit autophagy initiation by blocking the formation of the ULK1 complex [21]. The ULK1 complex recruits and activates the PI3KC3 complex, increasing PI3P levels, thereby facilitating its interaction with WIPI and DFCP1, which is necessary for nucleation/autophagosome formation [26,27,28].

During elongation and autophagosome formation, which involves nucleation, expansion, and maturation, the phagophore gives rise to the double-membrane autophagosome that requires two ubiquitin-like conjugate systems [29]. In the first ubiquitin-like system, the conjugation of the ATG12–ATG5 complex leads to the recruitment of the ATG16L1 protein, forming a multimeric complex known as ATG5–ATG12–ATG16 [30]. The second system is governed by LC3, encoded by the mammalian homolog of yeast, Atg8. LC3 is subjected to proteolytic cleavage at the C-terminus by Atg4B proteases, resulting in the formation of LC3-I [31]. Subsequently, through the combined action of the ATG7, ATG3, and ATG12–ATG5–ATG16 autophagy-related proteins, phosphatidylethanolamine (PE) is conjugated to LC3-I, promoting the generation of LC3-II and thus leading to its lipidation and localization at the outer and inner autophagosome membrane, aiding in the closure of the phagophore [31]. The presence of LC3-II on both the outer and inner membranes of autophagosomes is crucial for the effective operation of the autophagy pathway, which encompasses the creation, development, and final breakdown of autophagosomes [32]. During the fusion event, the lysosomal membranes involve the formation of autolysosomes by fusing with the outer membrane of autophagosomes [33]. 

Once autophagosome formation concludes, the LC3-II bound to the outer membrane undergoes cleavage, by Atg4, from phosphatidylethanolamine (PE) and is subsequently liberated back into the cytosol [33]. The fusion between the autophagosome and the lysosome is believed to necessitate the presence of diverse proteins, including integral lysosomal proteins (e.g., LAMP2), RAB proteins (e.g., RAB5 and RAB7), and SNARE complexes (e.g., VAMP8, STX17) [34]. Hydrolytic enzymes like cathepsins play a crucial role by degrading the contents of the autophagosome, which converts the cellular components into basic building blocks that are then transported from the lysosomal lumen into the cytosol for the synthesis of proteins and maintenance of cellular functions under conditions of nutrient deprivation and starvation [31] (Figure 1).

## 3. Formation and Biological Function of Cancer Stem Cells

Understanding the formation and biological characteristics of CSCs is crucial for diagnosing and treating tumors effectively. Despite advancements in tumor biology research, which have notably enhanced clinical diagnosis and cancer treatment in recent years, the unresolved issues of high recurrence and mortality rates are closely linked to the distinctive biological properties of CSCs. CSCs can self-renew and differentiate into multiple cancer cell types that make up the tumor, and they are responsible for tumor initiation, growth, and metastasis as well as resistance to therapy [35,36]. The ability for self-renewal is directly responsible for CSC tumorigenesis [37]. CSCs can undergo symmetrical cell division, giving rise to either two CSCs or one CSC and one daughter cell [38,39]. This facilitates the expansion of CSC populations, leading to extensive cell growth and ultimately contributing to the formation of tumors [38]. The self-renewal capacity of CSCs is validated via tumor serial transplantation, where CSCs extracted from a tumor are transplanted into a mouse model to assess their ability to initiate new tumor growth [40]. CSCs and normal stem cells share many similar regulatory signaling pathways, such as Sonic Hedgehog (Hh), Wnt/β-catenin, and Notch, which are crucial for their self-renewal properties [41,42]. Furthermore, signaling molecules such as PTEN also play key roles in regulating the growth of CSCs, and understanding the regulation of self-renewal is pivotal in gaining insight into tumorigenesis [43] (Figure 2). Additionally, the integration of distinct surface markers, functional assays for the ability to regenerate and differentiate, testing for tumorigenicity, gene expression analysis, and resistance to therapies offer a comprehensive set of tools for the identification and characterization of CSCs [44]. Knowledge of the CSC biology is essential because they can resist traditional treatments, which promotes aggressive tumor growth and unfavorable patient outcomes. Developing successful cancer medicines requires the addressing of the CSCs’ survival mechanisms.

Cancer stem cells are a minority group of malignant and oncogenic cells that can self-renew and are responsible for initiating and advancing tumor growth [45]. CSCs have crucial functions in the initiation, progression, resistance to cell death, resistance to therapy, and recurrence of tumors after treatment and eradication (Figure 3). Additionally, CSCs can generate heterogeneous cancer cell types. Discovering more effective markers for identifying CSCs and their heterogeneity is crucial, and methods such as proteomic profiling, gene expression analysis, next-generation sequencing, and genetic mutation analysis are frequently employed in cancer biomarker discovery [46,47]. Studies have revealed that both CD271^−^ and CD271^+^ melanoma cells were able to generate new tumors in immunodeficient NOD/SCID mice, while CD166 was proposed as a more dependable surface marker for lung CSCs compared to CD44, CD133, or EpCAM [48,49]. In addition, brain CSCs obtained from patients exhibited positivity for the markers CD133 and nestin, which are identical to the markers found on normal neuronal stem cells; however, certain cells lack surface markers indicative of differentiation [50]. CSCs can also undergo transdifferentiation into various multilineage cells, influencing tumorigenesis [51]. Another study discovered that renal CSCs differentiated into vascular endothelial cells (ECs) within the majority of tumors formed in SCID mice following the injection of human renal CSCs [52]. Furthermore, CSCs exhibiting differentiation into vascular ECs, thereby promoting angiogenesis, have been observed in several cancers including glioblastoma and liver cancer [53]. CSCs are strongly associated with tumor metastasis, responsible for tumor cells migrating to a distant site, and possess the ability to migrate to organs [54,55]. Numerous studies have indicated that metastatic cells exhibit a CSC phenotype, characterized by markers such as ALDH^+^ and CD44^+^CD24^−^ in breast cancer, CD26^+^ in colon cancer, and CD133^+^ in pancreatic cancer [56]. A study revealed a close association between CSCs and EMT, indicating EMT as a probable mechanism for tumor invasion and metastasis, while identifying CD44^+^/α2βhi1/CD133^+^ prostate cancer and CD133^+^/CXCR4^+^ pancreatic cancer cells as tumorigenic, highlighting the critical role of CSCs in tumor progression and metastasis [57]. Moreover, CSCs are also responsible for the drug resistance of the cancer cells by expressing multidrug resistance proteins, for example, different ATP-binding cassette (ABC) transporters, e.g., MRP1 (ABCC1) and (ABCG2), which prevent damage from drugs and therapies in cancers like leukemia [58]. Furthermore, in terms of plasticity that is responsible for giving rise to heterogenous cancer cells, CSCs represent a novel framework for comprehensively elucidating cancer initiation, progression, and the development of therapies to combat resistance to treatment [59].

## 4. Dual Role of Autophagy in the Maintenance of Cancer Stem Cell

Cancer stem cells represent a subpopulation of cancer cells with the capacity for self-renewal, differentiation, and the ability to drive tumor initiation, progression, and recurrence. Autophagy, a catabolic process that degrades and recycles cellular components, has been implicated in the regulation of CSCs, exhibiting both protective and destructive roles within the tumor microenvironment [60]. Autophagy promotes cell survival under stress conditions, such as nutrient deprivation, hypoxia, and therapeutic stress, which are common in tumor microenvironments. In CSCs, autophagy has been shown to support stemness and the resistance to chemotherapy and radiation therapy. For instance, studies have demonstrated that autophagy induction in CSCs leads to the activation of signaling pathways that promote stem cell maintenance and resistance to apoptosis [61]. This protective role is crucial for the survival of CSCs under adverse conditions, facilitating tumor relapse and metastasis after treatment. Autophagy enables CSCs to withstand harsh microenvironmental conditions by degrading damaged organelles and misfolded proteins, thereby preventing the accumulation of cellular debris, and supporting metabolic homeostasis. This process is critical for CSCs’ survival, as it provides essential metabolic substrates during nutrient deprivation, ensuring their growth and maintenance [62]. It has been observed that autophagy can drive the differentiation of CSCs, potentially limiting their tumorigenic potential. This process involves the degradation of damaged organelles and proteins, which can prevent the accumulation of cellular damage that contributes to genomic instability and cancer progression. The regulation of this process involves a complex network of signals including key transcription factors such as JNK, NF-Kappa B, HIF-1, and p53, which play roles in regulating genes important for autophagy [63]. The ability of CSCs to self-renew and differentiate is critical for tumor growth and metastasis. Autophagy has been shown to support the stemness of CSCs by regulating the signaling pathways crucial for maintaining their self-renewal capacity. For example, the activation of autophagy in glioma stem cells has been linked to the stabilization of hypoxia-inducible factors (HIFs), which are key regulators of the stem cell phenotype in hypoxic tumor niches [64] (Figure 4).

Programmed cell death 4 (PDCD4) acts as a tumor suppressor in lung cancer by reducing the expression of p62, a protein involved in autophagy, thereby inhibiting cell proliferation and tumorigenesis while promoting apoptosis. This interaction between PDCD4 and p62 highlights a potential therapeutic target for enhancing autophagic flux, which could lead to the suppression of lung cancer growth and the induction of cancer cell death. Understanding the PDCD4–p62 axis offers promising insights for developing novel strategies to improve treatment outcomes in lung cancer [65]. Although the suppression of PDCD4 can have notable impacts on the behavior of cancer cells, there is still no conclusive evidence that it specifically targets and eliminates CSCs. Available data indicate that inhibiting PDCD4 may typically enhance tumor aggressiveness and resistance, potentially negatively impacting both CSCs and non-CSCs [66]. Further study specifically designed to examine the impact of PDCD4 inhibition on CSCs is necessary to offer a conclusive response. This could involve carrying out research utilizing particular markers and assays that are exclusive to CSCs in order to differentiate the impact on CSCs from that of the overall tumor population [67]. Mitophagy, a selective autophagy process targeting mitochondria, is crucial for maintaining mitochondrial quality and homeostasis by removing damaged or dysfunctional mitochondria. A lack of mitophagy is associated with impaired mitochondrial function, which can contribute to the development and progression of various cancers by promoting tumorigenesis [68]. A deficiency in mitophagy is linked to mitochondrial dysfunction, the initiation of tumorigenesis, and the advancement of cancer across various types. Mitophagy serves diverse roles at different stages of cancer progression, including the following: it supports the metabolic requirements of normal cells and inhibits tumor formation during early tumorigenesis, while in the advanced stages, it boosts cell survival mechanisms, thereby facilitating cancer progression [69]. Autophagy plays a dual role in cancer progression, either inhibiting or facilitating tumor development based on the cancer type and its growth stage, influenced by its key role in mediating the complex interactions among the cancer cells, tumor environment, body’s tissues, and immune system. CSCs often reside in harsh microenvironments characterized by hypoxia, nutrient scarcity, and acidic conditions known to induce autophagy. Studies have shown that the basal rate of autophagy tends to be higher in CSCs compared to non-cancer stem cells, helping them respond to these stimuli and maintain their stemness under such adverse conditions [70].

Additionally, metabolic regulation plays a significant role in the maintenance and proliferation of CSCs. The metabolism of breast CSCs (BCSCs) is closely associated with a preference for anaerobic glucose metabolism, as indicated by the increased activities of the key enzymes in glycolysis. Researchers have successfully targeted and eliminated breast cancer stem cells by blocking important enzymes in the glycolytic pathway, including hexokinase 2 (HK2), phosphofructokinase (PFK), and lactate dehydrogenase A (LDHA) [71]. This process of selective death happens due to the reduced capacity of these cells to transition to other energy pathways in comparison with non-CSCs. Drugs such as 2-deoxy-D-glucose (2-DG) that hinder glycolysis have demonstrated the potential in preclinical investigations for specifically targeting breast CSCs [72]. The overall effectiveness of treatment can be improved by combining glycolysis inhibitors with other therapies, such as chemotherapy or radiotherapy, as this targets both CSCs and non-CSCs within the tumor [73]. In brief, the suppression of anaerobic glucose metabolism demonstrates the ability to specifically eliminate breast CSCs, indicating a promising approach in combating breast cancer [74]. Additional study and clinical trials are necessary to comprehensively comprehend and enhance these therapeutic methods. Metabolic adaptation is linked to a significant reduction in mitochondrial activity, further emphasizing the metabolic flexibility that supports CSC survival and proliferation [75]. Emerging evidence suggests that some CSCs also rely on oxidative phosphorylation (OXPHOS), indicating a metabolic heterogeneity among CSCs. This heterogeneity is a testament to the adaptability of CSCs, allowing them to survive and thrive in various microenvironmental conditions. Moreover, the metabolic status of CSCs is influenced by their location within the tumor, with oxygen availability (normoxia vs. hypoxia) playing a critical role in determining whether CSCs utilize glycolytic and/or oxidative metabolic pathways. Intratumorally, hypoxia, for example, promotes a glycolytic metabolism in CSCs through the upregulation of HIF-1α and its target glycolytic enzymes [76]. Exploring autophagy’s multifaceted role in CSCs offers a potential pathway for innovative cancer therapies. This approach is delicate due to autophagy’s dual function, that is to say it can either promote CSC survival or aid in their differentiation thus suppressing tumor growth. Identifying the optimal timing for inhibiting or inducing autophagy is key for effective treatment. Ongoing clinical trials aim to integrate autophagy modulators with standard cancer treatments, with the goal of overcoming resistance to therapy and preventing cancer recurrence [77]. Research into specific cancer types has revealed that targeting autophagy can be a viable strategy for cancer therapy, particularly by influencing the differentiation, survival, and proliferation of CSCs. For instance, salinomycin has been identified as an efficient agent against breast CSCs by suppressing autophagic flux [78]. Salinomycin has demonstrated a specific ability to target and eradicate CSCs, specifically in cases of breast cancer. Studies have shown that salinomycin is beneficial in decreasing the percentage of breast CSCs, which in turn hinders the growth and spread of tumors [79]. The specific focus on certain cells is one of the factors that has led to salinomycin being considered as a possible treatment for CSCs [78]. These cells are typically more resistant to standard chemotherapy and play a role in the recurrence and spread of tumors. Similarly, the combination of autophagy inhibitors with conventional cancer therapies, like chloroquine with gemcitabine in pancreatic CSCs, has shown improved drug sensitivity and increased susceptibility of CSCs to these treatments. Moreover, targeting autophagy has led to the sensitization of various CSC populations to traditional treatments, indicating the potential of autophagy modulation as a means to improve the efficacy of existing cancer therapies [15]. Research has demonstrated that interfering with autophagy in CSCs can result in their transformation into specialized cells, loss of their ability to regenerate, and heightened susceptibility to traditional treatments [80]. Suppressing autophagy can cause the build-up of malfunctioning mitochondria in CSCs, leading to heightened oxidative stress and cell demise [81]. Autophagy-targeting inhibitors can weaken these resistance mechanisms, rendering CSCs more vulnerable to apoptosis and other forms of cellular demise [60]. Autophagy inhibitors, such as chloroquine and hydroxychloroquine, have demonstrated a specific impact on CSC populations. This results in a reduction in tumor growth and the spread of cancer to other parts of the body in experimental models [82]. To summarize, CSCs have a greater susceptibility to autophagy inhibitors because they heavily depend on autophagy to sustain their stemness, longevity, and resistance to treatments. Autophagy inhibition is a possible therapeutic method for selectively targeting CSCs due to their heightened sensitivity.

## 5. Cancer Stem Cell Model of Tissue and Organoid Regeneration

Targeting CSCs holds promise for long-term cancer treatment, considering that stem/progenitor cells are the probable targets of tumorigenic mutations, given their longevity and inherent capabilities to accumulate mutations [83,84]. The transition of tumor cells into CSCs within primary tumors often leads to the acquisition of malignant traits and the development of distant metastases, driving tumor progression and treatment resistance [85,86,87]. Welch and Hurst outlined four key features that define the hallmarks of metastasis, encompassing motility, invasiveness, microenvironment modulation, plasticity, and colonization at secondary sites [88]. CSCs exhibit resistance to drug therapy through a variety of molecular processes, such as cellular plasticity, proficient DNA damage repair, the upregulation of genes associated with multi-drug resistance, and the inhibition of apoptosis. There is a clear association between CSCs and drug resistance in various types of human malignancies, such as breast, pancreatic, melanoma, leukemia, colorectal, and brain tumors. Autophagy is considered a crucial cellular and molecular mechanism in the development of resistance to chemotherapy drugs [2]. Due to the crucial significance of autophagy in the resistance of CSCs to chemotherapy, the focus on inhibiting this process has become a promising approach to making CSCs more responsive to chemotherapy. Interleukin 6 (IL-6) is a cytokine that potentially contributes to the development of both chronic and acute cancer stem cells (CSCs). The signaling pathway of IL-6 has been demonstrated to control the characteristics of stem cells that are responsible for cancer initiation and progression. Additionally, IL-6 signaling may also play a role in promoting the growth, invasion, and spread of different cancer types. Pharmacological substances that hinder the process of autophagy, such as chloroquine and hydroxychloroquine, have demonstrated effectiveness in laboratory models by reducing the survival of CSCs and increasing the destructive effects of chemotherapy [89]. The combination of autophagy inhibitors and conventional chemotherapeutic drugs has the potential to overcome resistance in CSCs and enhance treatment outcomes in cancer patients [90] (Figure 5). Nevertheless, the effectiveness of autophagy inhibitors in therapy may differ based on the specific type of tumor, microenvironmental conditions, and disease stage. Therefore, additional preclinical and clinical studies are required to confirm their usefulness in clinical settings.

The process of a metastatic cascade refers to the migration of cancerous cells from their primary site to other areas of the body, and it comprises a sequence of events, beginning with the emergence of metastatic cells, followed by uncontrolled proliferation, angiogenesis, motility, invasion, intravasation, dissemination, cellular arrest, vascular adhesion, and extravasation, culminating in colonization of the distant organs [88,91,92]. Understanding these processes at the cellular and tissue levels is essential for the development of accurate diagnostic tools and effective cancer therapies. While primary in vitro models for assessing cancer responses to systemic anticancer drugs have predominantly relied on 2D cultures, engineered 3D models have since emerged to overcome some limitations, facilitating tumor cell proliferation and differentiation [93]. These models include no scaffold, anchorage-independent, and scaffold-based systems, where 3D spheroids are formed and integrated into biomaterial scaffolds [94]. Variations exist in cell sources, cell preparation protocols, and the time required for 3D spheroid model formation [95,96,97]. One prominent approach for generating 3D culture models involves the induction of floating conditions for CSCs. The floating culture method operates on the nonadherent surfaces of plates, with three primary models as follows: the hanging drop method, the forced floating method, and the agitation-based method [98,99]. The hanging drop technique involves the creation of cell suspension droplets within well plates, which are then inverted, resulting in droplets hanging due to surface tension. In contrast, the forced floating method prevents cell adhesion to substrates by employing a non-adherent coating, thus enabling cells to float unrestrictedly. In the agitation method, cells in suspension are gently stirred to prevent attachment to substrates. A recent, cost-effective method for enriching CSC prostate cancer cells involves culturing with hydrophilic filter paper. This strategy encourages the spontaneous formation of tumor spheroids while enhancing the expression of CSC biomarkers. This is facilitated by the heightened hydrophilicity of cellulose fibers, which promotes cell aggregation within the confined spaces and niches between the fibers [100].

Initially, a soft hydrogel was used to create spheroid tissue models, facilitating the formation of new colonies by the differentiation of tumor and stromal cells in the absence of cell adhesion peptides [101,102]. Despite their potential, the limited throughput ability of cancer stem cells (CSCs), representing only about 0.4% of tumor cells, has hindered their broader application [103]. Multicellular spheroid tumor models have been created for investigating interactions between cells and their microenvironment. These models exhibit dense, spherical structures resembling in vivo tissues, facilitating the study of hypoxia-induced angiogenesis [104,105,106]. Compared to 2D cultures, spheroid models better replicate tumor conditions such as hypoxia-induced angiogenesis and reduced gas, nutrient, and drug exchange [107,108,109,110,111]. Studies investigating the effect of extracellular matrix (ECM) properties on glioma stem cell migratory responses in spheroid cultures revealed differences in migratory behaviors based on hydrogel porosity and stiffness [112]. Similarly, investigations into the interactions between macrophages and ovarian CSCs within hanging-drop-made spheroids demonstrated significant upregulation of key macrophage biomarkers by the CSCs [113]. Exploring drug responses in 3D culture models has revealed better representations of in vivo tumor conditions, including chemoresistance, metastasis, and recurrence, compared to 2D monolayers or dispersed 3D cultures [114]. Hepatocellular carcinoma, osteosarcoma, and lung carcinoma spheroid models enriched with CSC have potentially displayed aggressive tumor-initiating cell characteristics, such as drug resistance and high invasion capacity, reflecting tumor progression and metastasis [115,116,117]. Recent advancements in culture models have integrated microscale technologies, enabling compatibility with automated high-throughput screening [93].

Conventional 3D culture and organoid models face limitations in terms of monitoring migration processes and quantifying invasion, prompting researchers to explore microfluidic devices for more controlled experimentation [118]. Microfluidic devices, particularly when integrated with organs-on-chips, represent a promising avenue in cancer research, offering advantages such as reduced drug development time and cost, as well as mitigating ethical concerns associated with animal studies [119,120]. In contrast to traditional 2D cell cultures and animal models, 3D models such as organs-on-chips offer a closer resemblance to human pathophysiology [121]. Organs-on-chips are 3D organotypic devices capable of accommodating multiple cell types and providing flow and mechanical input [122]. Tumors-on-chips, a form of microfluidic device, simulate tumors by replicating their physiological conditions. They enable continuous the perfusion of nutrients and pharmaceutical testing [123]. By mimicking cancer behavior and offering insights into effective drug development, microfluidic models hold promise for enhancing our understanding of cancer biology [124]. Given that cancer cell motility is influenced by extracellular matrix components and the CSC phenomenon is implicated in therapeutic resistance, microfluidic models like tumors-on-chips could revolutionize CSC research [125,126,127,128,129,130].

## 6. Therapeutic Role of Autophagy in Cancer Stem Cell and Stem Cell Therapy

The therapeutic role of autophagy in cancer stem cell (CSC) therapy is increasingly recognized as a promising avenue in cancer treatment. CSCs, a subset of tumor cells with self-renewal and tumor-initiating properties, have been implicated in tumor relapse and drug resistance. Autophagy plays a crucial role in the maintenance and survival of CSCs under stress conditions [29]. Studies have demonstrated that targeting autophagy in CSCs can sensitize them to conventional chemotherapy and radiotherapy, thereby overcoming therapeutic resistance [131,132]. Notably, CSCs exhibit heightened levels of autophagy compared to non-stem cells, and inhibiting autophagy compromises CSC viability and tumorigenic potential. For instance, blocking autophagy in ovarian CSCs has shown synergistic effects with carboplatin treatment, underscoring its involvement in chemoresistance [133]. Moreover, the SOX2/β-catenin/Beclin1/autophagy signaling pathway has been implicated in the promotion of chemoresistance and stemness in colorectal cancer [134]. Furthermore, autophagy activation is linked to resistance against temozolomide and facilitates CSC plasticity [135]. Strategies targeting autophagy inhibition, such as salinomycin and chloroquine, demonstrate promise in sensitizing CSCs to conventional chemotherapy across various cancer types [135,136]. Additionally, modulating autophagy with agents like nicardipine and photodynamic therapy have emerged as potential approaches to heighten CSC sensitivity to therapy [137]. By targeting cancer stem-like cells and modulating the microenvironment of the tumor, resveratrol can inhibit the progression of lung cancer [138,139].

Various compounds and drugs have been investigated for their ability to modulate autophagy in CSCs, including inhibitors of signaling pathways such as Akt, mTOR, and Hedgehog signaling, etc., which are presented in Table 1. Additionally, pharmacological inhibitors targeting CSC-specific proteins like BMI-1 have shown promise in blocking CSC self-renewal and inhibiting tumor growth in preclinical models [87]. However, the heterogeneity of CSC populations and the complexity of autophagy regulation pose challenges in developing effective therapeutic strategies [87,140]. CSCs can exploit autophagy to evade cell death induced by therapeutic interventions, leading to treatment failure and disease recurrence. Studies have highlighted the role of autophagy in promoting survival and proliferation of CSCs under various stress conditions, thereby contributing to therapeutic resistance [134,141]. Furthermore, the crosstalk among autophagy, CSCs, and the tumor microenvironment play a critical role in modulating immune responses and promoting immune evasion in CSCs. Understanding the molecular mechanisms underlying autophagy-mediated therapy resistance in CSCs is essential for developing innovative therapeutic approaches targeting autophagy pathways. Combination therapies that simultaneously target autophagy and conventional cancer treatments hold promise for improving treatment outcomes and overcoming therapy resistance in CSCs [142]. However, further preclinical and clinical studies are warranted to validate the efficacy and safety of these combinatorial approaches and to translate them into clinical practice for better management of cancer.

Due to the crucial function of autophagy in maintaining and resisting therapy in cancer stem cells, the approach of targeting autophagy has become a potential option for eliminating CSCs and enhancing the effectiveness of stem cell therapy [158,159]. Pharmacological substances that regulate autophagy, such as chloroquine and hydroxychloroquine, have been studied as additional treatments to traditional cancer therapies to make cancer stem cells more susceptible to cell death caused by treatment [160]. In preclinical models of cancer, genetic interventions that specifically target important regulators of autophagy pathways, such as ATG genes and mTOR signaling, have demonstrated encouraging outcomes [161]. Autophagy is also crucial in controlling the destiny and operation of stem cells employed in regenerative medicine and cancer treatment [15]. Autophagy is responsible for maintaining the balance of stem cells by removing damaged organelles and protein aggregates [15]. This process helps to protect the ability of stem cells to renew themselves and differentiate into different cell types. Moreover, augmenting autophagy in stem cells can enhance their viability and integration after transplantation, hence amplifying the therapeutic efficacy of stem cell-based treatments [162]. Additional research endeavors are necessary to understand the intricate relationships among autophagy, cancer stem cells (CSCs), and stem cell treatment. These efforts are crucial for the application of these discoveries in a clinical setting for the advantage of individuals with cancer.

## 7. Limitation and Challenges of Autophagy in Cancer Stem Cell and Stem Cell Therapy

Recently, scientists have discovered interesting links between autophagy, CSCs, and stem cell therapies [143]. Although autophagy regulation has the potential for cancer treatment and stem cell-based therapies, its effectiveness and use in clinical settings are hindered by various limits and challenges. The dual involvement of autophagy in cancer is a major restriction. Autophagy can limit tumor initiation by eliminating damaged organelles and proteins, but it can also enable tumor persistence by providing nutrition and energy under stress. CSCs have increased autophagic activity, which helps them self-renew and withstand therapy [163]. Thus, targeting autophagy in CSCs demands a precise balance to disrupt its pro-tumorigenic actions without affecting its tumor-suppressive benefits [2]. The variability of CSC populations makes the therapeutic targeting of autophagy difficult. CSCs’ plasticity and ability to adapt to varied microenvironments cause subpopulations to have different autophagic activity [164]. A one-size-fits-all strategy to autophagy regulation may not eliminate CSCs or prevent tumor recurrence, requiring customized therapeutic regimens suited to the individual tumor’s autophagic characteristics. Autophagy-targeted therapeutics are also limited by tumor microenvironment complexity. Autophagic flux and CSC survival and proliferation are affected by solid tumor hypoxia, nutritional deficiency, and acidity [164]. CSC interactions with stromal cells, such as cancer-associated fibroblasts and immune cells, affect autophagy kinetics and treatment responses [15]. Thus, understanding the complex relationship between autophagy and the tumor microenvironment is essential for improving autophagy-based therapeutics.

However, different challenges must be overcome to fully utilize autophagy in stem cell therapy. The effects of autophagy modulation on stem cell destiny and function are unclear, generating concerns regarding the long-term safety and efficacy of autophagy-targeted methods. The time, duration, and dosage of autophagy modulators must be optimized to maximize the therapeutic advantages and minimize off-target effects and cytotoxicity. Autophagy has a role in stem cell therapy for cancer, regenerative medicine, and tissue engineering [165]. Stem cells have the innate autophagic machinery for self-renewal and differentiation [166]. Stem cell-based therapies are generally limited by low cell survival and engraftment rates in vivo. By boosting stem cell survival, engraftment, and regenerative capacity, autophagy regulation may improve their therapeutic potential. Therefore, autophagy has enormous potential as a cancer and stem cell therapy target, but its limitations must first be acknowledged and overcome to turn preclinical results into clinically effective treatments. Future research should elucidate the complicated relationship among autophagy, CSCs, and the tumor microenvironment, and optimize autophagy regulation techniques to improve stem cell-based therapies. These limitations must be overcome to properly utilize autophagy’s therapeutic potential in cancer treatment and regenerative medicine.

## 8. Conclusions

The manipulation of autophagy suggests a potentially fruitful approach to the targeting of cancer stem cells and the enhancement of the effectiveness of stem cell therapy [167]. Autophagy’s complex molecular pathways play a significant part in the regulation of CSCs, which in turn influences the cells’ ability to self-renew, differentiate, and survive during the process [15]. Furthermore, gaining a understanding of the biological roles of CSCs sheds light on their role in the regeneration of organs and tissues, which further emphasizes the necessity of targeting CSCs in the treatment of cancer and in regenerative medicine [168]. A dual role in the maintenance of CSCs is demonstrated by autophagy, which serves both as a survival strategy and as a potential vulnerability [169]. By utilizing this duality, therapeutic approaches could control autophagy to either promote the death of CSCs or make them more sensitive to traditional treatments. Both naturally occurring chemicals and synthesized medications have intriguing pathways for modifying autophagy and targeting CSCs; nevertheless, there are still hurdles to be overcome in maximizing their efficacy and limiting the effects that are not intended to be targeted. The utilization of autophagy for CSC-targeted therapeutics is not without its limits, even though it has the potential to be therapeutic. To achieve exact targeting, some hurdles must be overcome due to the intricacy of autophagy regulation and the context-dependent impacts it has. Furthermore, tailored approaches to therapy are required because of the variability of CSC populations and the microenvironments of tumors. Thus, the manipulation of autophagy is a viable technique for targeting CSCs and improving the outcomes of stem cell therapy in the treatment of cancer and regenerative medicine. To realize the full potential of autophagy regulation in clinical practice, it will be essential to continue the research efforts aimed at deciphering the complex mechanisms of autophagy and developing therapies that are specifically targeted.

## Figures and Tables

**Figure 1 cells-13-00958-f001:**
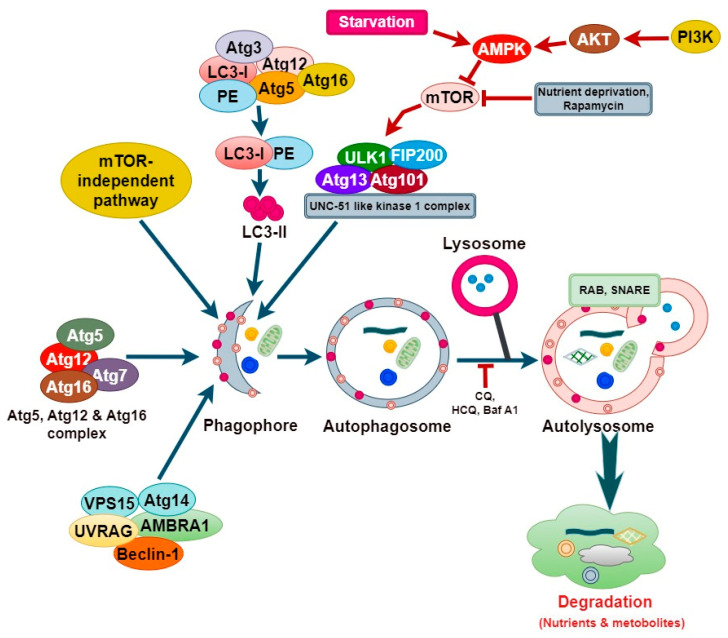
Molecular mechanism of autophagy. The autophagy signaling system is a tightly controlled, multi-step process essential for maintaining cellular balance. Initiation entails the activation of the ULK1 complex, nucleation necessitates the activity of the PI3K complex, elongation relies on ATG proteins, and completion involves the fusing of autophagosomes with lysosomes, which is mediated by the SNARE proteins and Rab GTPases. The regulation of these processes is crucial for providing the proper cellular responses to stress and the maintenance of cellular homeostasis.

**Figure 2 cells-13-00958-f002:**
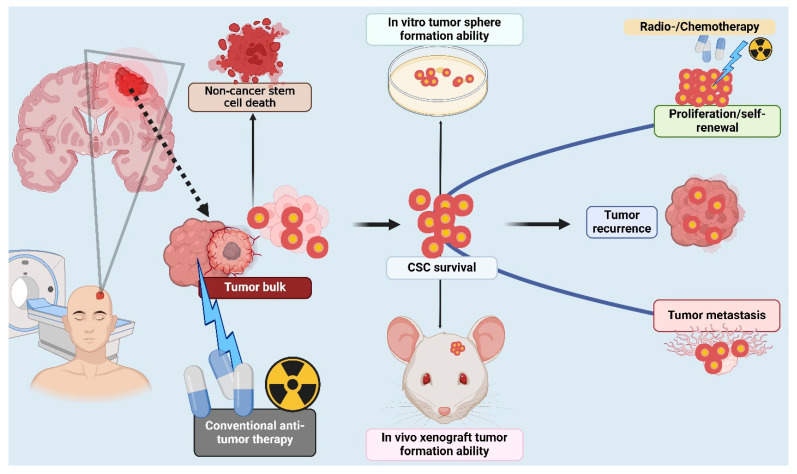
Formation of cancer stem cells. Cancer stem cells (CSCs) form through a complex interplay between the genetic and epigenetic alterations within tumors. Non-cancer stem cell death contributes to CSC survival, enhancing in vitro tumor sphere formation and in vivo xenograft growth. CSCs exhibit self-renewal and proliferation, driving tumor recurrence and metastasis. Environmental stress, hypoxia, and treatment resistance facilitate CSC formation, promoting tumor bulk heterogeneity. This figure was created with BioRender.com (accessed on 20 May 2024).

**Figure 3 cells-13-00958-f003:**
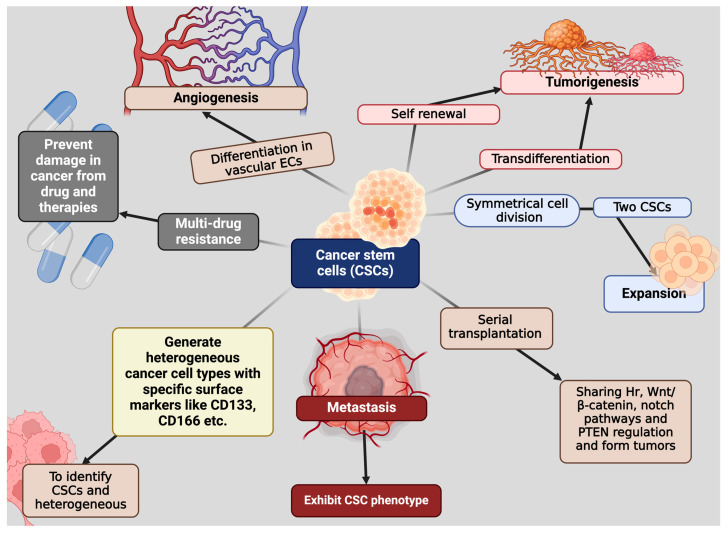
Biological function of cancer stem cells. CSCs play a crucial role in tumor formation and progression. They possess self-renewal capabilities, allowing them to regenerate and sustain tumors. These can divide symmetrically and form tumors as well. Additionally, CSCs have been implicated in metastasis, transdifferentiating, and transplantation. Moreover, they possess specific markers that help them to be recognized. Finally, drug resistance makes them uniquely different from normal stem cells. The unique attributes of CSCs differentiate them from regular stem cells and the majority of tumor cells, requiring the creation of specialized therapeutic approaches to efficiently target and eliminate CSCs in cancer therapy. All the biological characteristics of CSCs lead them to form tumors and subsequently cancer cells as well. This figure was created with BioRender.com (accessed on 20 May 2024).

**Figure 4 cells-13-00958-f004:**
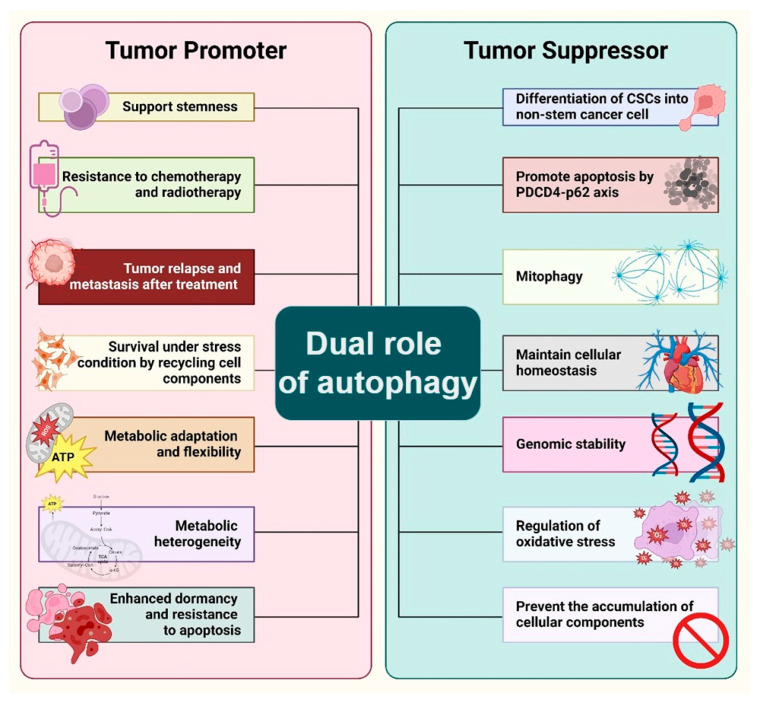
Dual role of autophagy in cancer stem cells. Autophagy has a multifaceted and paradoxical impact on development and differentiation, infection immunity and cancer suppression, as well as cell survival and homeostasis. On one hand, autophagy in CSCs has been linked to the stemness of CSCs, resistance to therapy, tumor relapse and metastasis, while conferring the ability to survive under stress conditions, metabolic flexibility and heterogeneity, and resistance to apoptosis on those cells that support tumor promotion and progression. On the other hand, differentiation of the CSCs into non-stem cancer cells, the promotion of apoptosis, mitophagy, cellular homeostasis, genomic stability, regulation of oxidative stress, and prevention of accumulation of cellular components give rise to suppressive activity via autophagy on the CSCs. This figure was created with BioRender.com (accessed on 20 May 2024).

**Figure 5 cells-13-00958-f005:**
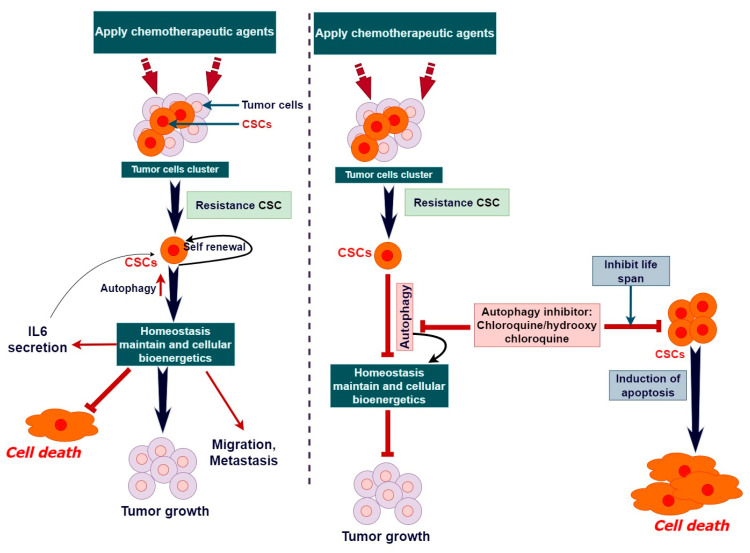
CSCs resistance to chemotherapy is mediated by autophagy signaling. The tumor consists of a heterogeneous cell population of clones derived from rapidly dividing tumor cells, together with a minority of cells known as cancer stem cells (CSCs) that could initiate tumor growth. Traditional treatments like radiotherapy and cytotoxic chemotherapy are successful in eliminating dividing cells. However, the autophagy-mediated cell survival mechanism protects tumor-initiating CSCs from being adversely affected by these therapies. Targeting autophagy in CSCs may potentially help overcome resistance and recurrence of the tumor after chemotherapy. This figure is derived from a previously published article by Rahman et al., 2020 [2].

**Table 1 cells-13-00958-t001:** An overview of the articles that have investigated autophagy in CSCs and outlined their key findings.

Aspect	Mechanism of Autophagy in CSCs	References
Therapeutic Role	Under stress, autophagy helps CSCs survive. Using autophagy to sensitize CSCs to conventional chemo and radiation can overcome treatment resistance.	[131]
CSCs Characteristics	CSCs have higher autophagy than non-stem cells, which helps them self-renew, initiate tumors, and resist therapy.	[143]
Chemoresistance	CSC viability and tumorigenicity can be reduced by autophagy inhibition. Blocking autophagy in ovarian CSCs enhances carboplatin therapy. The SOX2/β-catenin/Beclin1/autophagy pathway promotes colorectal cancer stemness and chemoresistance.	[144]
Drug Resistance	Activation of autophagy is associated with resistance to temozolomide and promotes the ability of cancer stem cells to change and adapt.	[145]
Therapeutic Strategies	Agents, such as salinomycin, chloroquine, nicardipine, and photodynamic treatment can sensitize CSCs to conventional chemotherapy by inhibiting autophagy. Resveratrol can slow lung cancer progression by targeting stem-like cells and altering the tumor microenvironment.	[146,147]
Pathway Inhibition	Drugs and chemicals that inhibit Akt, mTOR, and Hedgehog signaling pathways have been tested for their capacity to control CSC autophagy. Preclinical models indicate potential for CSC-specific protein inhibitors like BMI-1.	[148,149]
Challenges	Due to CSC heterogeneity and autophagy regulation complexity, developing effective treatments is tough. Therapy failure and disease recurrence occur because CSCs use autophagy to evade cell death from therapeutic interventions.	[150]
Immune Evasion	Autophagy, CSCs, and the tumor microenvironment interact to modulate immune responses and promote immune evasion in CSCs.	[151]
Combination Therapies	Combination medicines targeting autophagy and conventional cancer treatments may improve results and overcome therapeutic resistance in CSCs. More preclinical and clinical investigations are needed to confirm efficacy and safety.	[152]
Pharmacological Substances	In addition to typical cancer treatments, chloroquine and hydroxychloroquine have been explored to render CSCs more sensitive to cell death.	[153]
Genetic Interventions	Genetic therapies targeting autophagy regulators including ATG genes and mTOR signaling have demonstrated promising results in preclinical cancer models.	[154]
Regenerative Medicine	Regenerative medicine and cancer treatment depend on autophagy to control stem cell fate and function. It removes damaged organelles and protein aggregates to protect stem cells’ ability to replenish and differentiate.	[155,156]
Stem Cell Therapy	Autophagy increases stem cell survival and integration following transplantation, improving stem cell-based therapies. Research is needed to understand the complex interaction between autophagy, CSCs, and stem cell therapies.	[157]

## Data Availability

Not applicable.

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
