# Peer review of "Exploring Importance and Regulation of Autophagy in Cancer Stem Cells and Stem Cell-Based Therapies"

_cells, 2024, doi:10.3390/cells13110958_

Round 1
Reviewer 1 Report
Comments and Suggestions for Authors
As stated in their Introduction, "This review focuses on (a) comprehensive understanding of the complex relationship among autophagy, cancer stem cells, and stem cell therapy (that) presents promising opportunities for the progression of cancer therapeutics and regenerative medicine." p. 2, lines 98-101
The authors have done an excellent job at describing the relationship between autophagy and cancer stem cells (CSCs). The figures are perfect for teaching students at all levels. The review is easy to read and will definitely be helpful to anyone interested in autophagy and cancer. However, for the reader to appreciate the analysis provided by the authors of this review, the reader needs to understand what makes CSCs unique; how are CSCs distinguished from other cancer cells and from other types of stem cells. Otherwise, the road to developing therapeutic interventions targeted against CSCs will not be evident from this review.
I urge the authors to amplify and clarify the following points and present them as two separate sections. Points 1,2 and 3 could be combined into one section, and points 4 and 5 presented in another section.
(1) What, if anything, distinguishes CSCs from mammalian pluripotent embryonic stem cells (ESCs), induced pluripotent stem cells (iPSCs), and various tissue specific stem cells? Page 2, line70 states “Cancer stem cells (CSCs) are a separate type of cancer cell that could self-renew, is pluripotent, and is resistant to standard therapy [7].” This definition includes ESCs and iPSCs as well. All stem cells undergo self-renewal in vitro as well as in vivo, and both ESCs and iPSCs produce tumors whenever they find themselves at ectopic locations. Similarly, page 5, line 174 states that “Cancer stem cells are a minority group of malignant and oncogenic cells that could self-renew and are responsible for initiating and advancing tumor growth. Thus, teratocarcinomas derived from ESCs or iPSCs fit this definition. Should the reader consider ESCs and iPSCs as CSCs?
(2) Are CSCs cancer type specific? In other words, do breast CSCs only produce breast cancer? Cancer cell lines are cancer type specific; melanoma cell lines produce melanomas. To my knowledge, the direction of differentiation in pluripotent cells is determined by their environment. In contrast, tissue specific stem cells are programmed to differentiate into a single type of cell.
(3) How are CSCs distinguished from the plethora of 'cancer cell lines' that can 'self-renew' in vitro and produce a tumor when transferred to immune-compromised mice? Perhaps the critical distinction is that CSCs are tumorigenic populations capable of transferring human disease into immune-deficient mice and that can recapitulate the phenotype and morphology of the original patient tumors distinguish cancer stem cells from differentiated cancer cells. Please explain to the reader how CSCs are identified experimentally from other types of cancer cells.
(4) How might we selectively eliminate CSCs in the clinic? Section 4 addresses that question, but it does not answer it clearly. Does inhibition of PDCD4 selectively kill CSCs? Does inhibition of 'anaerobic glucose metabolism' selectively kill breast CSCs? Does salinomycin selectively eliminate breast CSCs or is it generally effective against all autophagy-dependent cancer cells?
(5) Are any aspects of CSCs more sensitive to inhibitors of autophagy than either ESCs, iPSCs or autophagy-dependent cancer cells that have been described in the literature?
Here are a few relevant publications that might be useful in addressing questions 4 and 5.
Nuclear magnetic resonance detects phosphoinositide 3-kinase/Akt-independent traits common to pluripotent murine embryonic stem cells and their malignant counterparts.
Romanska HM et al. Neoplasia. 2009 Dec;11(12):1301-8. PMID: 20019838
The tumorigenicity of diploid and aneuploid human pluripotent stem cells.
Blum B, Benvenisty N. Cell Cycle. 2009 Dec;8(23):3822-30. PMID: 19887907 Review.
Generation of pluripotent cancer-initiating cells from transformed bone marrow-derived cells.
Liu C et al., Cancer Lett. 2011 Apr 28;303(2):140-9. PMID: 21339043
Rediscovering pluripotency: from teratocarcinomas to embryonic stem cells.
Barbaric I, Harrison NJ. Int J Dev Biol. 2012;56(4):197-206. PMID: 22562197
Can pluripotent stem cells be used in cell-based therapy?
Picanço-Castro V, et al. Cell Reprogram. 2014 Apr;16(2):98-107. PMID: 24606201
Selective elimination of pluripotent stem cells by PIKfyve specific inhibitors
Chakraborty AR et al. Stem Cell Reports. 2022 Feb 8;17(2):397-412. PMID: 35063131
Endosome maturation links PI3Kα signaling to lysosome repopulation during basal autophagy.
Rodgers SJ et al. EMBO J. 2022 Oct 4;41(19):e110398. PMID: 35968799
Sequential conversion of PtdIns3P to PtdIns(3,5)P2 via endosome maturation couples nutrient signaling to lysosome reformation and basal autophagy.
Rodgers SJ et al. Autophagy. 2023 Apr;19(4):1365-1367. PMID: 36103410
Comments on the Quality of English Language
here are three trivial examples
"Exploring (the) Importance and Regulation of Autophagy in Cancer Stem Cells and Stem Cell-Based Therapies"
"This review focuses on (a) comprehensive understanding of the complex relationship among autophagy, cancer stem cells, and stem cell therapy (that) presents promising opportunities for the progression of cancer therapeutics and regenerative medicine." p. 2, lines 98-101
"3. Formation and biological function of cancer stem cell(s)" p4 line 148
Author Response
Review 1
First of all, we would like to express our sincere gratitude for the time and effort the reviewer had put into reviewing our manuscript.
As stated in their Introduction, "This review focuses on (a) comprehensive understanding of the complex relationship among autophagy, cancer stem cells, and stem cell therapy (that) presents promising opportunities for the progression of cancer therapeutics and regenerative medicine." p. 2, lines 98-101
>>Response: We are thankful to the reviewer for this complement.
The authors have done an excellent job at describing the relationship between autophagy and cancer stem cells (CSCs). The figures are perfect for teaching students at all levels. The review is easy to read and will definitely be helpful to anyone interested in autophagy and cancer. However, for the reader to appreciate the analysis provided by the authors of this review, the reader needs to understand what makes CSCs unique; how are CSCs distinguished from other cancer cells and from other types of stem cells. Otherwise, the road to developing therapeutic interventions targeted against CSCs will not be evident from this review.
>>Response: Thank you to reviewer wonderful comments.
I urge the authors to amplify and clarify the following points and present them as two separate sections. Points 1,2 and 3 could be combined into one section, and points 4 and 5 presented in another section.
>>Response: Thank you for reviewer suggestions. We organized the manuscript according to maintain consistency with title and content. Hope the reviewer considers the points.
(1) What, if anything, distinguishes CSCs from mammalian pluripotent embryonic stem cells (ESCs), induced pluripotent stem cells (iPSCs), and various tissue specific stem cells? Page 2, line70 states “Cancer stem cells (CSCs) are a separate type of cancer cell that could self-renew, is pluripotent, and is resistant to standard therapy [7].” This definition includes ESCs and iPSCs as well. All stem cells undergo self-renewal in vitro as well as in vivo, and both ESCs and iPSCs produce tumors whenever they find themselves at ectopic locations. Similarly, page 5, line 174 states that “Cancer stem cells are a minority group of malignant and oncogenic cells that could self-renew and are responsible for initiating and advancing tumor growth. Thus, teratocarcinomas derived from ESCs or iPSCs fit this definition. Should the reader consider ESCs and iPSCs as CSCs?
>>Response:
Cancer Stem Cells (CSCs): CSCs are a specific subpopulation of cancer cells with the ability to self-renew and differentiate into various cell types within the tumor. They are thought to be responsible for tumor initiation, progression, metastasis, and recurrence.
Embryonic Stem Cells (ESCs): ESCs are derived from the inner cell mass of blastocysts and are inherently pluripotent, meaning they can differentiate into all cell types of the body.
Induced Pluripotent Stem Cells (iPSCs): iPSCs are generated by reprogramming somatic cells to a pluripotent state using specific transcription factors. Like ESCs, iPSCs can differentiate into any cell type.
Distinguishing CSCs from ESCs and iPSCs:
Unlike ESCs and iPSCs, CSCs are directly associated with cancer. ESCs and iPSCs are not inherently cancerous but can give rise to teratomas, which are non-malignant tumors. CSCs specifically contribute to the malignancy of tumors, driving cancer growth, metastasis, and resistance to treatment. ESCs and iPSCs do not exhibit these behaviors unless they undergo malignant transformation, which is not their inherent characteristic.
The reader should not consider ESCs and iPSCs as CSCs despite some overlapping properties such as self-renewal and tumor formation capabilities. The key differences lie in their origin, role in cancer, and behavior in therapeutic contexts. CSCs are inherently malignant, contributing to cancer progression and therapeutic resistance, which is not a characteristic of ESCs and iPSCs unless they undergo abnormal transformations.
Yes, Page 2, line70 description of cancer stem cells (CSCs) is correct.
Yes, the statement in page 5, line 174 is correct. Cancer stem cells (CSCs) are indeed a minority population within a tumor.
(2) Are CSCs cancer type specific? In other words, do breast CSCs only produce breast cancer? Cancer cell lines are cancer type specific; melanoma cell lines produce melanomas. To my knowledge, the direction of differentiation in pluripotent cells is determined by their environment. In contrast, tissue specific stem cells are programmed to differentiate into a single type of cell.
>>Response: Cancer stem cells (CSCs) are generally considered to be cancer type-specific. This means that CSCs originating from a particular type of cancer, such as breast cancer, will typically give rise to the same type of cancer, in this case, breast cancer. This specificity is due to several factors such as Tissue of Origin, Genetic and Epigenetic Profiles, Microenvironment Influence, and Hierarchical Organization of Tumors. Therefore, similar to how tissue-specific stem cells are programmed to differentiate into a particular type of cell, CSCs are typically programmed to produce cancer cells of the same type from which they originated. This is why breast CSCs will produce breast cancer, melanoma CSCs will produce melanoma, and so on.
(3) How are CSCs distinguished from the plethora of 'cancer cell lines' that can 'self-renew' in vitro and produce a tumor when transferred to immune-compromised mice? Perhaps the critical distinction is that CSCs are tumorigenic populations capable of transferring human disease into immune-deficient mice and that can recapitulate the phenotype and morphology of the original patient tumors distinguish cancer stem cells from differentiated cancer cells. Please explain to the reader how CSCs are identified experimentally from other types of cancer cells.
>>Response: Cancer stem cells (CSCs) are a subpopulation of cancer cells with distinct properties that set them apart from the bulk of cancer cell lines that can also self-renew and form tumors. CSCs are distinguished from the general population of cancer cell lines by their unique markers, high tumorigenic potential, ability to self-renew and differentiate, resistance to therapies, quiescent state, reliance on specific stem cell pathways, and results from functional assays. These characteristics make CSCs a critical target for developing more effective cancer treatments aimed at eradicating the root of cancer persistence and recurrence. We added this information page 4 line 169-176
(4) How might we selectively eliminate CSCs in the clinic? Section 4 addresses that question, but it does not answer it clearly. Does inhibition of PDCD4 selectively kill CSCs? Does inhibition of 'anaerobic glucose metabolism' selectively kill breast CSCs? Does salinomycin selectively eliminate breast CSCs or is it generally effective against all autophagy-dependent cancer cells?
>>Response: Implementing strategies in clinical settings requires a multidisciplinary approach, integrating insights from oncology, molecular biology, immunology, and pharmacology to effectively target and eliminate CSCs in the clinic.
Each question we respond to on page 7 line 262-269, page 8 line 291-303, and page 8 line 323-328.
(5) Are any aspects of CSCs more sensitive to inhibitors of autophagy than ESCs, iPSCs or autophagy-dependent cancer cells that have been described in the literature?
>>Response: CSCs exhibit a higher sensitivity to autophagy inhibitors due to their reliance on autophagy for maintaining stemness, survival, and resistance to therapies and sensitivity makes autophagy inhibition a promising therapeutic strategy for targeting CSCs specifically.
More response are in page 8 line 334-346
Here are a few relevant publications that might be useful in addressing questions 4 and 5.
Nuclear magnetic resonance detects phosphoinositide 3-kinase/Akt-independent traits common to pluripotent murine embryonic stem cells and their malignant counterparts.
Romanska HM et al. Neoplasia. 2009 Dec;11(12):1301-8. PMID: 20019838
The tumorigenicity of diploid and aneuploid human pluripotent stem cells.
Blum B, Benvenisty N. Cell Cycle. 2009 Dec;8(23):3822-30. PMID: 19887907 Review.
Generation of pluripotent cancer-initiating cells from transformed bone marrow-derived cells.
Liu C et al., Cancer Lett. 2011 Apr 28;303(2):140-9. PMID: 21339043
Rediscovering pluripotency: from teratocarcinomas to embryonic stem cells.
Barbaric I, Harrison NJ. Int J Dev Biol. 2012;56(4):197-206. PMID: 22562197
Can pluripotent stem cells be used in cell-based therapy?
Picanço-Castro V, et al. Cell Reprogram. 2014 Apr;16(2):98-107. PMID: 24606201
Selective elimination of pluripotent stem cells by PIKfyve specific inhibitors
Chakraborty AR et al. Stem Cell Reports. 2022 Feb 8;17(2):397-412. PMID: 35063131
Endosome maturation links PI3Kα signaling to lysosome repopulation during basal autophagy.
Rodgers SJ et al. EMBO J. 2022 Oct 4;41(19):e110398. PMID: 35968799
Sequential conversion of PtdIns3P to PtdIns(3,5)P2 via endosome maturation couples nutrient signaling to lysosome reformation and basal autophagy.
Rodgers SJ et al. Autophagy. 2023 Apr;19(4):1365-1367. PMID: 36103410
here are three trivial examples
"Exploring (the) Importance and Regulation of Autophagy in Cancer Stem Cells and Stem Cell-Based Therapies"
"This review focuses on (a) comprehensive understanding of the complex relationship among autophagy, cancer stem cells, and stem cell therapy (that) presents promising opportunities for the progression of cancer therapeutics and regenerative medicine." p. 2, lines 98-101
"3. Formation and biological function of cancer stem cell(s)" p4 line 148
>>Response: We are grateful to the reviewer for suggestion of these articles.
Reviewer 2 Report
Comments and Suggestions for Authors
This is a well written manuscript regarding the role of autophagy in CSCs. It is relevant to the autophagy and cancer field and provides an overview of recent literature on the subject. However, some modifications need to be made before the manuscript can be accepted for publication.
My specific points are as follow:
Line 128, it says dilapidation, please change to lipidation
Line 129, LC3II is localized both to the outer and inner autophagosomal membranes, please correct. See Kabeya et al. EMBO J, 2000.
Line 137, it says play a role a crucial role, please correct.
Figure 1, it says Phagospore formation, please correct.
Figure 1, although mTOR responds to AMPK, AMPK can directly phosphorylate and activate ULK1, please correct and incorporate the ULK1 box with the UNC-51 like kinase 1, they are the same protein. mTOR is one of the main autophagy inhibitors in response to nutrient sufficiency, particularly amino acids, it should be depicted as such and not only as an AMPK downstream target. Please see Kim J et al. Nature cell biology, 2011.
Figure 1 legend, it says it forms autolysosome and perform autophagy, autophagy starts since the autophagome is nucleated, please correct.
Figure 2. The figure legend does not describe what is shown in the figure. Also, the authors state that cells undergo de-differentiation and acquire the features of stem cells. While this is true, another possibility is that mutations happen in the normal stem cell, please consider changing this legend.
Figure 3, it says tumerogenesis, please correct. It also says heterogenetic, please change to heterogeneous
Figure 3, legend. It says drug resistance makes them uniquely different from normal stem cells. It is proposed that both normal and CSCs are particularly resistant to chemotherapy or radiotherapy since they divide slower than the tumor bulk, please explain.
Line 187, it says transdifferentiating, please change to transdifferentiation
Figure 4, legend, it says, Autophagy plays a complex and contradictory role in bipolar disorder. Please correct.
Figure 5 is confusing. The authors need to check the figure legend for grammar and style correction. The title says: CSCs resist chemotherapy mediated autophagy, do they mean through autophagy, by autophagy? Please check. It is also not clear what the green circle below autophagy means. On the figure, why do the authors center on IL6? Is that particularly important for CSCs? They secrete other cytokines/chemokines…
Also for figure 5, how is life span related to CSC biology? Life span inhibition is not related to cell death.
Also for figure 5, it says resistance to CSC, do the authors mean resistant CSCs?
Please check this figure since it is very confusing.
Section 5 mentions several points that have been mentioned previously on section 3, they should incorporate the repeated points on section 3 to avoid repetition.
In order to summarize their manuscript, the authors need to add a table incorporating the studies where autophagy has been studied in CSCs and their main findings.
Comments on the Quality of English LanguageEnglish is fine, but some errors need to be corrected as specified on my review report.
Author Response
Review 2
This is a well written manuscript regarding the role of autophagy in CSCs. It is relevant to the autophagy and cancer field and provides an overview of recent literature on the subject. However, some modifications need to be made before the manuscript can be accepted for publication.
>>Response: First, we would like to express our sincere gratitude for the time and effort the reviewer had put into reviewing our manuscript.
My specific points are as follow:
Line 128, it says dilapidation, please change to lipidation
>>Response: Changed page 3 line 128.
Line 129, LC3II is localized both to the outer and inner autophagosomal membranes, please correct. See Kabeya et al. EMBO J, 2000.
>>Response: We modified accordingly with reference in page 3 line 130-132.
Line 137, it says play a role a crucial role, please correct.
>>Response: we corrected in page 3 line 140.
Figure 1, it says Phagospore formation, please correct.
>>Response: We corrected Phagosphore to “Phagophore” in page 4 figure 1.
Figure 1, although mTOR responds to AMPK, AMPK can directly phosphorylate and activate ULK1, please correct and incorporate the ULK1 box with the UNC-51 like kinase 1, they are the same protein. mTOR is one of the main autophagy inhibitors in response to nutrient sufficiency, particularly amino acids, it should be depicted as such and not only as an AMPK downstream target. Please see Kim J et al. Nature cell biology, 2011.
>>Response: Thank to the reviewer for this wonderful suggestion. We modified the figure accordingly to the reviewer comments in page 4 figure 1.
Figure 1 legend, it says it forms autolysosome and perform autophagy, autophagy starts since the autophagome is nucleated, please correct.
>>Response: We corrected autophagy starts with autophagosome is nucleated and finally degradation in page 4 line 150.
Figure 2. The figure legend does not describe what is shown in the figure. Also, the authors state that cells undergo de-differentiation and acquire the features of stem cells. While this is true, another possibility is that mutations happen in the normal stem cell, please consider changing this legend.
>>Response: We modified and corrected figure legend 2 in page 5 line 177-181.
Figure 3, it says tumerogenesis, please correct. It also says heterogenetic, please change to heterogeneous
>>Response: We changed tumerogenesis to “Tumorigenesis” and heterogenetic to “heterogeneous” in page 6 figure 3.
Figure 3, legend. It says drug resistance makes them uniquely different from normal stem cells. It is proposed that both normal and CSCs are particularly resistant to chemotherapy or radiotherapy since they divide slower than the tumor bulk, please explain.
>>Response: We explain to add the following sentences: CSCs exhibit intrinsic resistance mechanisms that are not typically present in normal stem cells or the bulk of the tumor cells. These characteristics make them distinct from normal stem cells and the bulk of tumor cells, necessitating the development of specialized treatment strategies to effectively target and eradicate CSCs in cancer therapy in figure 3 legend page 6.
Line 187, it says transdifferentiating, please change to transdifferentiation
>>Response: We corrected transdifferentiating to “transdifferentiation “in page 5 line 196.
Figure 4, legend, it says, Autophagy plays a complex and contradictory role in bipolar disorder. Please correct.
>>Response: Corrected, Autophagy has a multifaceted and paradoxical impact on bipolar disorder in figure 4 legend page 9 line 348.
Figure 5 is confusing. The authors need to check the figure legend for grammar and style correction. The title says: CSCs resist chemotherapy mediated autophagy, do they mean through autophagy, by autophagy? Please check. It is also not clear what the green circle below autophagy means. On the figure, why do the authors center on IL6? Is that particularly important for CSCs? They secrete other cytokines/chemokines…
>>Response: We corrected, modified and made new figure 5 with legends in page 10 and 11 line 384-391 .
Also for figure 5, how is life span related to CSC biology? Life span inhibition is not related to cell death.
>>Response: We modified and corrected figure 5 legends in page 11.
Also for figure 5, it says resistance to CSC, do the authors mean resistant CSCs?
>>Response: We modified and corrected figure 5 legends in page 11.
Please check this figure since it is very confusing.
>>Response: We changed figure 5 page 10.
Section 5 mentions several points that have been mentioned previously on section 3, they should incorporate the repeated points on section 3 to avoid repetition.
>>Response: We deleted repeated sentence from section 5.
In order to summarize their manuscript, the authors need to add a table incorporating the studies where autophagy has been studied in CSCs and their main findings.
>>Response: As per suggestion, we added a table in the section “6. Therapeutic role of autophagy in cancer stem cell and stem cell therapy” in CSCs and their main findings in page 12 and 13.
English is fine, but some errors need to be corrected as specified on my review report.
>>Response: We check overall English grammar and typo.
Round 2
Reviewer 2 Report
Comments and Suggestions for Authors
The authors have responded most of my previous comments. However, some changes need to be applied before the manuscript can be accepted for publication.
My specific comments:
Figure 1 is still confusing. Class III PI3K is downstream to ULK1 and nutrient deprivation regulates ULK1 both by modulating AMPK and by not activating mTOR. It is not clear why are there arrows from nutrient deprivation to Class III PI3K and to ULK1. How would nutrient deprivation directly regulate these targets? Since this manuscript is about autophagy, the autophagic pathway needs to be clearly depicted in this figure. The authors are referred to great reviews in the area in order to clearly understand and depict the authophagic pathway in this figure. See for example Klionsky DJ, et al, EMBOJ, Autophagy in major human diseases, 2021.
Figure 4 legend. Bipolar disorder is a neurological disease. Please do not use this term to avoid confusion.
The table is a good addition to the manuscript, but it needs to be referenced. Please add a third column with the references supporting the idea in each table line.
Comments on the Quality of English LanguageEnglish is OK, my specific suggestions are mentioned in my comments.
Author Response
The authors have responded most of my previous comments. However, some changes need to be applied before the manuscript can be accepted for publication.
First of all, we would like to express our sincere gratitude for the time and effort the reviewer had put into reviewing our manuscript.
My specific comments:
Figure 1 is still confusing. Class III PI3K is downstream to ULK1 and nutrient deprivation regulates ULK1 both by modulating AMPK and by not activating mTOR. It is not clear why are there arrows from nutrient deprivation to Class III PI3K and to ULK1. How would nutrient deprivation directly regulate these targets? Since this manuscript is about autophagy, the autophagic pathway needs to be clearly depicted in this figure. The authors are referred to great reviews in the area in order to clearly understand and depict the authophagic pathway in this figure. See for example Klionsky DJ, et al, EMBOJ, Autophagy in major human diseases, 2021.
>>Response: We make a new figure 1 and wrote new figure legends in page 4 line 146-151.
Figure 4 legend. Bipolar disorder is a neurological disease. Please do not use this term to avoid confusion.
>>Response: We deleted all bipolar and added dual role in figure 4 and legend in page 9.
The table is a good addition to the manuscript, but it needs to be referenced. Please add a third column with the references supporting the idea in each table line.
>>Response: We added respective references in all contents in page 14 table 1.